# Job satisfaction among healthcare workers in Ghana and Kenya during the COVID-19 pandemic: Role of perceived preparedness, stress, and burnout

**Patience A. Afulani**[1,2,3]*, **Jerry John Nutor**[4], **Pascal Agbadi**[5], **Akua O. Gyamerah**[6], **Joseph Musana**[7], **Raymond A. Aborigo**[8], **Osamuedeme Odiase**[2], **Monica Getahun**[2], **Linnet Ongeri**[9], **Hawa Malechi**[10], **Moses Obimbo Madadi**[11], **Benedicta Arhinful**[12], **Ann Marie Kelly**[13], **John Koku Awoonor-Williams**[14]

1 Department of Epidemiology & Biostatistics, University of California, San Francisco, San Francisco, California, United States of America, 2 Institute for Global Health Sciences, University of California, San Francisco, San Francisco, California, United States of America, 3 Department of Obstetrics, Gynecology & Reproductive Sciences, University of California, San Francisco, San Francisco, California, United States of America, 4 Department of Family Health Care Nursing, University of California, San Francisco, San Francisco, California, United States of America, 5 Department of Sociology and Social Policy, Lingnan University, Tuen Mun, Hong Kong, SAR, 6 Department of Community Health Systems, University of California, San Francisco, San Francisco, California, United States of America, 7 Department of Obstetrics and Gynecology, The Aga Khan University Hospital, Nairobi, Kenya, 8 Navrongo Health Research Centre, Navrongo, Ghana, 9 Centre for Clinical Research, Kenya Medical Research Institute, Nairobi, Kenya, 10 Tamale Teaching Hospital, Tamale, Ghana, 11 Department of Human Anatomy, Obstetrics and Gynecology, The University of Nairobi, Nairobi, Kenya, 12 School of Public Health, Johns Hopkins University, Baltimore, MD, United States of America, 13 Sidney Kimmel Medical College, Philadelphia, PA, United States of America, 14 Policy Planning Monitoring and Evaluation Division, Ghana Health Service, Accra, Ghana

* patience.afulani@ucsf.edu

**Data Availability Statement:** The data used for the analysis is included in the Supporting information files.

## Abstract

The COVID-19 pandemic has affected job satisfaction among healthcare workers; yet this has not been empirically examined in sub-Saharan Africa (SSA). We addressed this gap by examining job satisfaction and associated factors among healthcare workers in Ghana and Kenya during the COVID-19 pandemic. We conducted a cross-sectional study with healthcare workers (N = 1012). The two phased data collection included: (1) survey data collected in Ghana from April 17 to May 31, 2020, and (2) survey data collected in Ghana and Kenya from November 9, 2020, to March 8, 2021. We utilized a quantitative measure of job satisfaction, as well as validated psychosocial measures of perceived preparedness, stress, and burnout; and conducted descriptive, bivariable, and multivariable analysis using ordered logistic regression. We found high levels of job dissatisfaction (38.1%), low perceived preparedness (62.2%), stress (70.5%), and burnout (69.4%) among providers. High perceived preparedness was positively associated with higher job satisfaction (adjusted proportional odds ratio (APOR) = 2.83, CI [1.66,4.84]); while high stress and burnout were associated with lower job satisfaction (APOR = 0.18, CI [0.09,0.37] and APOR = 0.38, CI [0.252,0.583] for high stress and burnout respectively). Other factors positively associated with job satisfaction included prior job satisfaction, perceived appreciation from management, and

**Funding:** This study was funded by the University of California, San Francisco COVID-19 Related Rapid Research Pilot Initiative (Grant number #2016796), awarded to PAA and JJN. The funders had no role in the data collection, analysis, or decision to publish.

**Competing interests:** The authors declare that they have no competing interests.

perceived communication from management. Fear of infection was negatively associated with job satisfaction. The COVID-19 pandemic has negatively impacted job satisfaction among healthcare workers. Inadequate preparedness, stress, and burnout are significant contributing factors. Given the already strained healthcare system and low morale among healthcare workers in SSA, efforts are needed to increase preparedness, better manage stress and burnout, and improve job satisfaction, especially during the pandemic.

## Introduction

Healthcare workers (HCWs) are facing unprecedented professional challenges during the COVID-19 pandemic [1–3]. Since the beginning of the pandemic, numerous studies have documented prevalent anxiety, depression, and distress among frontline HCWs. However, a very limited number of studies have examined the ways in which job satisfaction has been impacted. Among the published studies, however, the findings are very clear: HCWs worldwide are largely dissatisfied with their jobs in the context of the COVID-19 pandemic [1–3].

Job satisfaction, defined as a collection of feelings and beliefs that people have about their current jobs, or an emotional response defining the degree to which people like their jobs, may range from extreme satisfaction to extreme dissatisfaction [4, 5]. Job satisfaction has been shown to impact performance, commitment, absenteeism, retention, and turnover rates [6–10]. It also has a bidirectional relationship with stress and burnout [11, 12]. Thus among HCWs, job satisfaction has important implications for quality of care and health outcomes [11–14].

Studies on job satisfaction among HCWs in sub-Saharan Africa (SSA) prior to the pandemic showed high levels of dissatisfaction [8, 15–18]. But job satisfaction in SSA has not been examined in the context of the rapidly-evolving COVID-19 pandemic [11]. Factors associated with job satisfaction are both personal and job-related. These include sociodemographic characteristics such as age, gender, and education [11, 19], and work-related and institutional factors such as type of occupation, professional status, years of service, workload, salary and rewards, work environment, adequacy of resources, job security, appreciation of efforts, relationship to other staff and managers, and career development [11, 17, 20].

The rapidly evolving and unprecedented pandemic is undoubtedly a contributing predictor of HCW job satisfaction. The pressure on HCWs, along with the many challenges they face, including higher risk of infection [21], fear of being infected and infecting family and loved ones [22, 23], heavier workloads [24, 25], mental health burden [26–28], inadequate personal protective equipment (PPE), lack of support [23, 29], and disturbances in work-life balance have contributed to decreased job satisfaction. This unrelenting burden of the pandemic on HCWs has contributed to the risk of HCW shortages. In a survey of nurses in the United States, it was found that three in five nurses intended to leave the workforce due to their experience with the COVID-19 pandemic, including the lack of PPE, intense workloads, and other physical and mental stressors [30].

HCW attrition and its associated impacts are more severe in low-and middle-income countries and further threaten the stability of fragile health systems. Further, HCW attrition due to job dissatisfaction in SSA are some of the highest in the world, underscoring the urgency and need for studies examining job satisfaction amidst the COVID-19 pandemic [7, 8, 31, 32]. One potential unexplored predictor is providers' perceived preparedness to respond to the pandemic. In a prior study in Ghana, we found that HCWs did not feel prepared for the pandemic

response [33], contributing to high stress and burnout [34], which has implications for job satisfaction.

To inform workforce development efforts that will impact both current and future pandemic response in SSA, this study examined job satisfaction among HCWs during the COVID-19 pandemic and the factors associated with it. Our primary objective was to examine job satisfaction among HCWs in Ghana and Kenya during the COVID-19 pandemic—focusing on the role of providers' perceived preparedness. We hypothesized that higher perceived preparedness will be associated with higher job satisfaction. In addition, given the established relationship between stress, burnout, and job satisfaction; and between preparedness, stress, and burnout [34], we also examined if the relationship between preparedness and satisfaction was mediated by stress and burnout. Further, we examined differences in job satisfaction and key predictors at different time points during the pandemic and across Ghana and Kenya.

## Methods

### Setting

Ghana and Kenya have had similar trends in the progression of the COVID-19 pandemic. Ghana, having recorded its first two cases on March 12, 2020, has reported a total of 94,824 cases (2.6% of all cases in Africa), 790 reported deaths, and administered 1.23 million vaccine doses as of June 17, 2021 [35, 36]. Kenya's first confirmed case was recorded on March 13, 2020. Since then, cases have increased exponentially with sharp increases during a second wave in late 2020 and a third wave from March to May 2021. As of June 17, 2021, in Kenya, there are 176,622 confirmed cases (4.8% of all cases in Africa), 3428 deaths, and 942,344 administered vaccine doses [35, 36]. Both countries have constrained healthcare systems; Ghana has less than one hospital bed per 1,000 people and an estimated 1.8 medical doctors and 42 nurses/midwives per 10,000 [37–39]. Similarly, Kenya has about one hospital bed per 1,000 people and an estimated 1.5 medical doctors and 8.3 nurses/midwives per 10,000 [40, 41].

Public health measures, including periodic lockdowns and movement restrictions, curfews, restriction of business hours, and closure of places of worship, have been implemented in both countries to curb the devastating effects of the virus both on the health of the population and the economy [42]. However, the increasing number of cases within previously overburdened health systems is a significant concern for HCWs, especially in the absence of widespread public compliance with preventative measures. Moreover, inadequate PPE and preparation further compounds HCWs' concern of being infected with COVID-19. These concerns sparked threats of strikes by nurses and doctors in Ghana earlier in the pandemic [43, 44]. Further, as of April 2021, HCWs made up about 10% of Ghana's cases, raising concerns about the disproportionate impact of the pandemic on frontline HCWs. Similarly in Kenya, facilities and HCWs have been overwhelmed by the number of patients needing COVID-19 care, causing stress and mental distress [45]. HCWs in Kenya have also raised concerns about preparedness, including lack of adequate PPE, fear of infection, morbidity and mortality among frontline HCWs, economic hardships, lack of adequate health insurance coverage, and adverse mental health [46]. These concerns have led to periodic threats of and actual strikes by different cadres of HCWs in Kenya [47].

Kenya and Ghana were chosen for this study because of the similarities in the COVID-19 response in both countries and our existing collaborations in the two countries. But despite the similarities, it is important to note that the differences in the number of HCW-patient ratios in the two countries may influence job satisfaction among HCW in the two countries before and during the COVID-19 pandemic.

## Data collection

We conducted a cross-sectional study in Ghana and Kenya with HCWs including nurses, physicians, and allied HCWs (medical laboratory, pharmacy, public health, and other healthcare staff). The two-phased data collection included: (1) data collected in Ghana from April 17 to May 31, 2020, and (2) data collected in Ghana and Kenya from November 9, 2020 to March 8, 2021. We recruited HCWs online through various platforms (WhatsApp, Facebook, mails, and direct messaging) utilizing HCW professional networks, contacts, and peer distribution strategies. Participants were invited to complete a self-administered online *Qualtrics* survey. HCWs working in either Ghana or Kenya were eligible to participate. While two versions of the survey were fielded, questions were similar across countries. No incentive was provided for Phase 1 study data collection and in Kenya. In Phase 2, Ghanaian survey respondents were entered to win a raffle for 200 Ghana Cedis (~$20 USD) for five randomly selected people.

The surveys were conducted in English and were pretested with 10 HCWs each in Ghana and Kenya. Questions covered basic demographics, perceived preparedness for the COVID-19 pandemic, job satisfaction, stress, burnout, and other questions relevant to the pandemic response. HCWs were provided brief consent language prior to the survey start and had the option of skipping questions. A total of 945 and 717 HCWs started the survey (i.e., answered the first question in the survey) in Ghana in phase 1 and phase 2, respectively; and 258 HCWs started in phase 2 in Kenya. Additional study methods are described elsewhere [33, 34].

## Ethical approval

The study received ethical approvals from the Institutional Review Boards of the University of California San Francisco; Navrongo Health Research Centre, Ghana; and the Aga Khan University, Kenya. All participants provided informed consent. Information about the study was provided on the first page of the online questionnaire. Respondents provided informed consent by proceeding to take the survey.

## Measures

**Dependent variable: Job satisfaction during the pandemic.** Job satisfaction was measured by a single question; "In general, how satisfied are you with your job now?" Response options were very dissatisfied, dissatisfied, satisfied, and very satisfied.

**Primary independent variable: Perceived preparedness.** A 15-item scale capturing personal, facility, and psychological preparedness for prevention, diagnoses, management, and education regarding COVID-19 captured perceived preparedness to respond to COVID-19. Each question had response options ranging from 0 (not prepared at all) to 3 (very prepared), with options for "I don't know about this" (4), and "not applicable to my role" (5). The scale development and validation process in Ghana has previously been published [33].

**Potential mediators: Perceived stress and burnout.** The 10-item Cohen perceived stress scale measure perceived stress vis-à-vis people's feelings and thoughts in the past month [48]. Questions captured how nervous or stressed, unpredictable, uncontrollable, and overloaded respondents found their lives, with response options to each question ranging from 0 (never) to 4 (very often). The 14-item Shirom-Melamed Burnout measure (SMBM) captured feelings at work in the past month [49], across three-domains of burnout: physical fatigue, emotional exhaustion, and cognitive weariness and with response options ranging from 1 (never or almost never) to 7 (always or almost always). Additional methods have also been previously described [34]. The psychometric properties for both measures have been previously assessed in SSA [34, 50].

**Other independent variables.** Other independent variables included job satisfaction prior to pandemic ("in general, how satisfied were you with your job before the COVID-19 crisis?"); feeling of appreciation and communication from management; family support; fear of contracting COVID-19; provider and facility characteristics; time points in the pandemic; and country (S1 Questionnaire).

## Data analysis

All analyses were performed using *STATA Version 16*. Initial analyses included data cleaning and sample selection. Study samples were built around the dependent and three main independent variables in addition to the correlates in each model and thus, consisted of respondents who answered all questions on job satisfaction, perceived preparedness, stress, burnout, and selected correlates (S1 Dataset). The sample with perceived preparedness as the main independent variable in the model consisted of n = 1009 participants. The study samples of the stress and burnout models were n = 897 and n = 887, respectively.

Factor analyses and inter-item reliability analyses were performed on complete data on all items of the three main scales (preparedness, stress, and burnout). All three scales had good construct validity, with all items in each scale loading at greater than three on one dominant factor with eigen-values greater than one. The scales also had good internal consistency with Cronbach alpha of 0.92 for preparedness, 0.82 for stress, and 0.95 for burnout. These statistics are identical with the ones reported in previous studies using these three scales [33, 34]. Items in each scale were recoded so that higher scores would mean higher preparedness, higher stress, and burnout.

The next analyses involved the derivation of the summative scores for each scale. The coded response options for the perceived preparedness scale ranged from 0 to 3 by recoding 4 (I don't know about this) to 0 (not at all prepared) and 5 (not applicable to my role) to 2 (prepared). The summative preparedness scores range from 0–45. The scores were categorized as follows: less than 15 or "not at all prepared"; 15 to 29 or "somewhat prepared," and $\geq$30 or "prepared" [33]. The summative perceived stress scores range from 0 to 40. Scores of 0–13 were considered low stress, 14–26 moderate stress, and 27–40 high stress [48]. The summative burnout scores range from 14–98—rescaled to 1–7 by dividing by the total number of items for ease of comparison with sub-domains. Scores of $\leq$2.0 were considered no burnout, 2–3.74 moderate burnout, and $\geq$3.75 as high burnout [51].

Frequency and percentages were used to describe the study samples, stratifying them by the data collection phase and country (Ghana Phase 1, Ghana Phase 2, and Kenya Phase 2). We then explored changes in job satisfaction and key predictors over time and by country. Given that the dependent variable, job-satisfaction was ordered, we used the ordered logistic regression technique to conduct the test of association between the dependent and the independent variables and to fit the multivariable models. The respondents were distributed across 44 clusters (urban and rural areas of twenty regions across both countries). Given the hierarchical nature of the data, we performed a two-level multilevel analysis to account for the contribution of the clusters in explaining the variability in the outcome. The intra-cluster correlation coefficient (ICC) of the models were zero and the coefficients of the regressors in both the single level and the multilevel models were identical, suggesting that the level two variable had little to no contribution to the model. Thus, we present only the single level models. The Variance Inflation Factor (VIF) was used to check for the assumptions of multicollinearity among the independent variables, and no violations were observed.

Mediation analysis were performed to assess whether the relationship between perceived preparedness and current job satisfaction was mediated by either burnout or stress using the

'khb' package in STATA [52]. Results from the mediation analysis guided the testing of other models, as explained in the results section. Given the regression coefficients of the association between perceived job satisfaction prior to the pandemic and current job satisfaction, sensitivity analyses were performed involving fitting the models with and without the prior job satisfaction variable. The adjusted pseudo-R squared were used to assess if the inclusion or exclusion of prior job satisfaction had any substantial effect on the models (Table B in S1 Table).

## Results

### Descriptive results

Table 1 shows the description of the participants with the combined sample and further stratified by the data collection phases: Ghana phase 1 and 2 and Kenya Phase 2. The demographic characteristics are quite diverse: about half of the combined sample were female and 30–39 years old, and most (71%) were currently married. About a quarter (23%) were doctors (20.2%, 25.5%, and 28.9% respectively for Ghana phase1, Ghana phase 2, and Kenya) and six out of ten were in the nursing field (63.2%, 55.9%, and 44.5% respectively for Ghana phase1, Ghana phase 2, and Kenya). A quarter of the sample worked in a teaching hospital, six out of ten worked in other government health facilities, and 13% worked in a private/mission owned health facility.

About 38% of HCWs were currently dissatisfied with their work (7.4% were very dissatisfied and 30.7% were dissatisfied). About six out of ten were currently satisfied with their work (52.3% were satisfied and 9.6% were very satisfied), compared to about 85% being satisfied prior to the pandemic (59.3% were satisfied and 25.6% were very satisfied). Healthcare workers in Ghana were more likely to be dissatisfied with their jobs during the early phase of the pandemic (8.8% were very dissatisfied and 34.7% were dissatisfied in Phase 1 data collection) compared to the later phases (4.2% were very dissatisfied, and 28.7% were dissatisfied in Phase 2). There was no significant difference in HCW satisfaction levels for Ghana and Kenya during the later phases (12.5% were very dissatisfied and 22.7% dissatisfied among Kenya sample).

Approximately 51% and 38% of HCWs felt a little prepared and prepared, respectively. HCWs felt less prepared in Ghana in Phase 1 (27.7%) than in Phase 2 (49.5%) in Ghana and in Kenya (38.3%) (Table A in S1 Table). About two-thirds (65%) of the healthcare workers were experiencing moderate stress, while 5% reported experiencing high stress. Nearly half of the workers (46%) reported low burnout while 23% had high burnout. The proportion of HCWs reporting both high burnout and high stress was greater in the Kenyan sample (high burnout: 37.3%; high stress: 11.3%) compared to Ghana Phase 1 (high burnout: 19.9%; high stress: 4.3%) and Phase 2 (high burnout: 22.6%; high stress: 4.4%). The distributions of other independent variables are shown in Table 1.

### Bivariable results

Table 2 shows the bivariable associations between each independent and control variable and current job satisfaction using cross-tabulations and ordered unadjusted logistic regressions. Perceived preparedness was positively associated with current job satisfaction, while perceived stress and burnout were negatively associated with current job satisfaction. Job satisfaction prior to the covid-19 pandemic, perceived appreciation from management, perceived communication from management, and perceived support from family were positively associated with current job satisfaction. Fear of COVID-19 infection was negatively associated with current job satisfaction. The following factors were not statistically significantly associated with current job satisfaction in the bivariable analyses: gender, age, marital status, type of healthcare provider, and the type of healthcare facility.

**Table 1. Summary statistics of study variables.**

| | Combined sample (N = 1012) | Ghana Phase 1 (N = 476) | Ghana Phase 2 (N = 408) | Kenya (N = 128) |
|---|---|---|---|---|
| **Study Variables** | n (%), unless otherwise indicated | | | |
| **Gender** | | | | |
| Male | 497 (49.11) | 236 (49.58) | 207 (50.74) | 54 (42.19) |
| Female | 515 (50.89) | 240 (50.42) | 201 (49.26) | 74 (57.81) |
| **Age** | | | | |
| < 30 | 264 (26.09) | 131 (27.52) | 104 (25.49) | 29 (22.66) |
| 30–39 | 527 (52.08) | 266 (55.88) | 216 (52.94) | 45 (35.16) |
| 40+ | 221 (21.84) | 79 (16.60) | 88 (21.57) | 54 (42.19) |
| **Marital status** | | | | |
| Single | 262 (25.89) | 122 (25.63) | 117 (28.68) | 23 (17.97) |
| Currently married | 718 (70.95) | 343 (72.06) | 278 (68.14) | 97 (75.78) |
| Formerly married (divorced/widowed) | 32 (3.16) | 11 (2.31) | 13 (3.19) | 8 (6.25) |
| **Position** | | | | |
| Doctor | 237 (23.42) | 96 (20.17) | 104 (25.49) | 37 (28.91) |
| Nurse/related | 586 (57.91) | 301 (63.24) | 228 (55.88) | 57 (44.53) |
| Other | 189 (18.68) | 79 (16.60) | 76 (18.63) | 34 (26.56) |
| **Health facility type** | | | | |
| Teaching hospital | 263 (25.99) | 125 (26.26) | 122 (29.90) | 16 (12.50) |
| Other government facility | 614 (60.67) | 280 (58.82) | 249 (61.03) | 85 (66.41) |
| Private/Missions | 135 (13.34) | 71 (9.07) | 37 (9.07) | 27 (21.09) |
| *Dependent variable* | | | | |
| **Current job satisfaction** | | | | |
| Very dissatisfied | 75 (7.41) | 42 (8.82) | 17 (4.17) | 16 (12.50) |
| Dissatisfied | 311 (30.73) | 165 (34.66) | 117 (28.68) | 29 (22.66) |
| Satisfied | 529 (52.27) | 234 (49.16) | 224 (54.90) | 71 (55.47) |
| Very satisfied | 97 (9.58) | 35 (7.35) | 50 (12.25) | 12 (9.38) |
| *Independent variables* | | | | |
| **Perceived preparedness** [M (SD); Min., Max.] | 26.10 (8.96); 3, 45 | 24.05 (8.79); 3, 44 | 28.46 (8.45); 5, 45 | 26.23 (9.37); 4, 45 |
| Not at all prepared | 109 (10.77) | 73 (15.34) | 23 (5.64) | 13 (10.16) |
| A little prepared | 520 (51.38) | 271 (56.93) | 183 (44.85) | 66 (51.56) |
| Prepared | 383 (37.85) | 132 (27.73) | 202 (49.51) | 49 (38.28) |
| **Perceived stress** [M (SD); Min., Max.] | 16.78 (6.15);0, 38 | 16.34 (5.84); 2, 34 | 16.57 (6.25); 0, 36 | 19.00 (6.47); 2, 38 |
| Low stress | 265 (29.48) | 130 (31.10) | 111 (30.33) | 24 (20.87) |
| Moderate stress | 587 (65.29) | 270 (64.59) | 239 (65.30) | 78 (67.83) |
| High stress | 47 (5.23) | 18 (4.31) | 16 (4.37) | 13 (11.30) |
| **Perceived burnout** [M (SD); Min., Max.] | 39.13 (16.34);14, 98 | 37.38 (15.55); 14, 98 | 39.29 (16.20); 14, 98 | 45.21 (18.31); 14, 95 |
| No burnout | 272 (30.60) | 137 (33.25) | 109 (29.70) | 26 (23.64) |
| Low burnout | 411 (46.23) | 193 (46.84) | 175 (47.68) | 43 (39.09) |
| High burnout | 206 (23.17) | 82 (19.90) | 83 (22.62) | 41 (37.27) |
| *Control variables* | | | | |
| **Job satisfaction prior to pandemic** | | | | |
| Very dissatisfied | 24 (2.37) | 11 (2.31) | 10 (2.45) | 3 (2.34) |
| Dissatisfied | 126 (12.75) | 61 (12.82) | 49 (12.01) | 19 (14.84) |
| Satisfied | 600 (59.29) | 277 (58.19) | 255 (62.50) | 68 (53.13) |
| Very satisfied | 259 (25.59) | 127 (26.68) | 94 (23.04) | 38 (29.69) |
| **Perceived appreciation from management** | | | | |
| Not at all appreciative | 122 (12.06) | 66 (13.87) | 39 (9.56) | 17 (13.28) |

*(Continued)*

**Table 1.** (Continued)

|  | Combined sample (N = 1012) | Ghana Phase 1 (N = 476) | Ghana Phase 2 (N = 408) | Kenya (N = 128) |
|---|---|---|---|---|
| Somewhat appreciative | 416 (41.11) | 192 (40.34) | 173 (42.40) | 51 (39.84) |
| Appreciative | 373 (36.86) | 179 (37.61) | 148 (36.27) | 46 (35.94) |
| Very appreciative | 101 (9.98) | 39 (8.19) | 48 (11.76) | 14 (10.94) |
| **Fear of infection** |  |  |  |  |
| Not fearful | 172 (17.00) | 59 (12.39) | 100 (24.51) | 13 (10.16) |
| A little fearful | 408 (40.32) | 202 (42.44) | 178 (43.63) | 28 (21.88) |
| Fearful | 242 (23.91) | 116 (24.37) | 75 (18.38) | 51 (39.84) |
| Very fearful | 190 (18.77) | 99 (20.80) | 55 (13.48) | 36 (28.13) |
| **Management communication** |  |  |  |  |
| Very poor | 98 (9.69) | 59 (12.42) | 31 (7.60) | 8 (6.25) |
| Poor | 303 (29.97) | 152 (32.00) | 118 (28.92) | 33 (25.78) |
| Good | 522 (51.63) | 223 (46.95) | 225 (55.15) | 74 (57.81) |
| Very good | 88 (8.70) | 41 (8.63) | 34 (8.33) | 13 (10.16) |
| **Support from family** |  |  |  |  |
| Not at all supportive | 55 (5.45) | 29 (6.12) | 24 (5.88) | 2 (1.56) |
| A little supportive | 226 (22.38) | 127 (26.79) | 80 (19.61) | 19 (14.84) |
| Supportive | 455 (45.05) | 208 (43.88) | 191 (46.81) | 56 (43.75) |
| Very supportive | 274 (27.13) | 110 (23.21) | 113 (27.70) | 51 (39.84) |

## Multivariable results

Three separate multivariable models were fitted with current job satisfaction as the dependent variable (Table 3). Each model had one of these variables as the main independent variable: perceived preparedness, burnout, and stress. The models were fitted because of the theoretical assumption that either stress or burnout mediates the relationship between perceived preparedness and job satisfaction. Burnout may mediate the effect of perceived stress. The results showed that each of the main independent variables were significantly associated with current job satisfaction after controlling for socio-demographic factors.

In model I, HCWs who were a little prepared [AOR = 1.679, 95% CI: 1.052, 2.678] or prepared [AOR = 2.831, 95% CI: 1.657, 4.838] had higher odds of being currently satisfied with their job compared to those who were not prepared at all. In model II, HCWs who experienced moderate [AOR = 0.513, 95% CI: 0.369, 0.715] or high stress [AOR = 0.180, 95% CI: 0.0868, 0.371] had lower odds of being currently satisfied with their job compared to those who experienced low stress. In model III, HCWs who experienced low [AOR = 0.662, 95% CI: 0.474, 0.923] or high burnout [AOR = 0.383, 95% CI: 0.252, 0.583] had lower odds of being currently satisfied with their job compared to those who experienced no burnout.

In each of the multivariable models, job satisfaction prior to the covid-19 pandemic, perceived appreciation from management, and perceived communication from management were positively associated with current job satisfaction, while fear of the COVID-19 infection was negatively associated with current job satisfaction. Phase 1 providers in Ghana also had lower odds of satisfaction than providers in phase 2 in both Ghana and Kenya. Perceived family support was significant only in model I. Marital status was significant in models II and III, with currently married HCWs being associated with lower odds of current job satisfaction.

The potential mediating effect of stress or burnout in the relationship between perceived preparedness and current job satisfaction were assessed by including stress and burnout separately in the model with preparedness (controlling for other sociodemographic factors) (Table 4). In model II and III where perceived stress and burnout were controlled for, only the

**Table 2. Bivariable association between key independent and control variables and current job satisfaction.**

| Study Variables | N, % Very Dissatisfied | N, % Dissatisfied | N, % satisfied | N, % Very satisfied | POR [95% CI] |
|---|---|---|---|---|---|
| *Independent variable* | | | | | |
| **Perceived preparedness** | | | | | |
| Not at all prepared | 25, 22.94 | 56, 51.38 | 25, 22.94 | 3, 2.75 | Ref. |
| A little prepared | 37, 7.12 | 187, 35.96 | 276, 53.08 | 20, 3.85 | 3.578*** [2.398,5.340] |
| Prepared | 13, 3.39 | 68, 17.75 | 228, 59.53 | 74, 19.32 | 11.88*** [7.668,18.39] |
| **Perceived stress** | | | | | |
| Low stress | 10, 3.77 | 50, 18.87 | 151, 56.98 | 54, 20.38 | Ref. |
| Moderate stress | 39, 6.64 | 208, 35.43 | 304, 51.79 | 36, 6.13 | 0.356*** [0.264,0.479] |
| High stress | 20, 42.55 | 12, 25.53 | 14, 29.79 | 1, 2.13 | 0.0647*** [0.0337,0.125] |
| **Perceived burnout** | | | | | |
| No burnout | 8, 2.94 | 58, 21.32 | 162, 59.56 | 44, 16.18 | Ref. |
| Low burnout | 22, 5.35 | 129, 31.38 | 222, 54.01 | 38, 9.25 | 0.548*** [0.405,0.741] |
| High burnout | 35, 16.99 | 81, 39.32 | 81, 39.32 | 9, 4.37 | 0.226*** [0.158,0.325] |
| *Control variables* | | | | | |
| **Job satisfaction prior to pandemic** | | | | | |
| Very dissatisfied | 13, 54.17 | 7, 29.17 | 2, 8.33 | 2, 8.33 | Ref. |
| Dissatisfied | 25, 19.38 | 91, 70.54 | 13, 10.08 | 0, 0.00 | 2.634* [1.083,6.409] |
| Satisfied | 29, 4.83 | 172, 28.67 | 384, 64.00 | 15, 2.50 | 21.34*** [8.898,51.17] |
| Very satisfied | 8, 3.09 | 41, 15.83 | 130, 50.19 | 80, 30.89 | 99.26*** [39.56,249.0] |
| **Perceived management appreciation** | | | | | |
| Not at all appreciative | 40, 32.79 | 50, 40.98 | 28, 22.95 | 4, 3.28 | Ref. |
| Somewhat appreciative | 26, 6.25 | 169, 40.63 | 206, 49.52 | 15, 3.61 | 4.389*** [2.903,6.636] |
| Appreciative | 7, 1.88 | 76, 20.38 | 238, 63.81 | 52, 13.94 | 13.59*** [8.751,21.10] |
| Very appreciative | 2, 1.98 | 16, 15.84 | 57, 56.44 | 26, 25.74 | 23.76*** [13.39,42.15] |
| **Fear of infection** | | | | | |
| Not fearful | 7, 4.07 | 38, 22.09 | 95, 55.23 | 32, 18.60 | Ref. |
| A little fearful | 14, 3.43 | 122, 29.90 | 237, 58.09 | 35, 8.58 | 0.610** [0.428,0.870] |
| Fearful | 13, 5.37 | 81, 33.47 | 132, 54.55 | 16, 6.61 | 0.482*** [0.328,0.709] |
| Very fearful | 41, 21.58 | 70, 36.84 | 65, 34.21 | 14, 7.37 | 0.199*** [0.130,0.302] |
| **Management communication** | | | | | |
| Very poor | 24, 24.49 | 52, 53.06 | 22, 22.45 | 0, 0.00 | Ref. |
| Poor | 30, 9.90 | 123, 40.59 | 130, 42.90 | 20, 6.60 | 3.347*** [2.160,5.186] |
| Good | 18, 3.45 | 121, 23.18 | 328, 62.84 | 55, 10.54 | 8.708*** [5.672,13.37] |
| Very good | 3, 3.41 | 14, 15.91 | 49, 55.68 | 22, 25.00 | 17.75*** [9.781,32.20] |
| **Support from family** | | | | | |
| Not at all supportive | 10, 18.18 | 28, 50.91 | 13, 23.64 | 4, 7.27 | Ref. |
| A little supportive | 23, 10.18 | 100, 44.25 | 95, 42.04 | 8, 3.54 | 1.701 [0.976,2.964] |
| Supportive | 21, 4.62 | 126, 27.69 | 268, 58.90 | 40, 8.79 | 3.987*** [2.334,6.810] |
| Very supportive | 21, 7.66 | 55, 20.07 | 153, 55.84 | 45, 16.42 | 5.492*** [3.129,9.639] |
| **Gender** | | | | | |
| Male | 31, 6.24 | 141, 28.37 | 281, 56.54 | 44, 8.85 | Ref. |
| Female | 44, 8.54 | 170, 33.01 | 248, 48.16 | 53, 10.29 | 0.811 [0.641,1.026] |
| **Age** | | | | | |
| < 30 | 20, 7.58 | 88, 33.33 | 132, 50.00 | 24, 9.09 | Ref. |
| 30–39 | 39, 7.40 | 165, 31.31 | 276, 52.37 | 47, 8.92 | 1.067 [0.806,1.412] |
| 40+ | 16, 7.24 | 58, 26.24 | 121, 54.75 | 26, 11.76 | 1.340 [0.951,1.888] |
| **Marital status** | | | | | |

*(Continued)*

**Table 2.** (Continued)

| Study Variables | N, % Very Dissatisfied | N, % Dissatisfied | N, % satisfied | N, % Very satisfied | POR [95% CI] |
|---|---|---|---|---|---|
| Single | 15, 5.73 | 82, 31.30 | 141, 53.82 | 24, 9.16 | Ref. |
| Currently married | 56, 7.80 | 215, 29.94 | 378, 52.65 | 69, 9.61 | 0.961 [0.735,1.257] |
| Formerly married (divorced/widowed) | 4, 12.50 | 14, 43.75 | 10, 31.25 | 4, 12.50 | 0.536 [0.265,1.085] |
| **Position** | | | | | |
| Doctor | 19, 8.02 | 68, 28.69 | 131, 55.27 | 19, 8.02 | Ref. |
| Nurse/related | 43, 7.34 | 197, 33.62 | 288, 49.15 | 58, 9.90 | 0.926 [0.695,1.234] |
| Other | 13, 6.88 | 46, 24.34 | 110, 58.20 | 20, 10.58 | 1.273 [0.883,1.835] |
| **Health facility type** | | | | | |
| Teaching hospital | 20, 7.60 | 90, 34.22 | 128, 48.67 | 25, 9.51 | Ref. |
| Other government facility | 41, 6.68 | 187, 30.46 | 329, 53.38 | 57, 9.28 | 1.160 [0.882,1.527] |
| Private/Missions | 14, 10.37 | 34, 25.19 | 72, 53.33 | 15, 11.11 | 1.209 [0.809,1.804] |
| **Country/phases** | | | | | |
| Ghana Phase 1 | 42, 8.82 | 165, 34.66 | 234, 49.16 | 35, 7.35 | 0.609*** [0.472,0.785] |
| Ghana Phase 2 | 17, 4.17 | 117, 28.68 | 224, 54.90 | 50, 12.25 | Ref. |
| Kenya | 16, 12.50 | 29, 22.66 | 71, 55.47 | 12, 9.38 | 0.778 [0.529,1.145] |

Exponentiated coefficients; 95% confidence intervals in brackets.

* p<0.05,

** p<0.01,

*** p<0.001.

prepared category of the perceived preparedness variable was significantly associated with current job satisfaction. In model II, accounting for perceived stress, HCWs who were prepared had higher odds of being satisfied compared to those who were not at all prepared [APOR = 2.776, 95% CI: 1.533, 5.029]. In the same model, accounting for perceived preparedness, moderate [AOR = 0.529, 95% CI: 0.379, 0.738] and high stress [APOR = 0.183, 95% CI: 0.0882, 0.379] were negatively associated with current job satisfaction. In model III, accounting for burnout, HCWs who were prepared had higher odds of being satisfied compared to those who were not at all prepared [APOR = 2.638, 95% CI: 1.441, 4.829]. In the same model, accounting for perceived preparedness, low [APOR = 0.712, 95% CI: 0.509, 0.997] and high burnout [AOR = 0.424, 95% CI: .278, 0.648] were negatively associated with current job satisfaction. However, as shown in Table 4, the change in magnitude of the coefficients from model 1 to II and III was small and the potential mediation effect using the Khb method was not significant (Table C in S1 Table), suggesting significant independent effects of perceived preparedness, stress, and burnout on job satisfaction. The observations made regarding the control variables in the models in Table 4 were identical to that in Table 3.

The multivariable analyses stratified by country and phase are shown in the supplemental tables (Tables D to G in S1 Table). The directions of the associations are in general consistent with the results from the combined sample. However, the confidence intervals are much wider for the Kenya sample, with higher p-values, because of the smaller sample size. For example, HCWs in both countries who felt prepared, had lower perceived stress, lower burnout, greater perceived appreciation from management, and perceived greater support from their family were more likely to have higher job satisfaction than those who felt less prepared, had higher perceived stress, higher burnout, felt less appreciated by their management, and less support from their family respectively (Table G in S1 Table). However, the effects of preparedness, perceived stress, and family support are not statistically significant for the Kenya sample when job satisfaction prior to the pandemic is added to the model (Table E in S1 Table).

**Table 3. Multivariable model showing the association between three IVs [perceived preparedness, burnout, and stress] and current job satisfaction, controlling for sociodemographic factors.**

| | Model I | Model II | Model III |
|---|---|---|---|
| | IV: Perceived Preparedness | IV: Perceived Stress | IV: Perceived Burnout |
| Study Variables | | | |
| *Independent variable* | APOR [95% CI] | APOR [95% CI] | APOR [95% CI] |
| **Perceived preparedness** | | | |
| Not at all prepared | Ref. | | |
| A little prepared | 1.679* [1.052,2.678] | | |
| Prepared | 2.831*** [1.657,4.838] | | |
| **Perceived stress** | | | |
| Low stress | | Ref. | |
| Moderate stress | | 0.513*** [0.369,0.715] | |
| High stress | | 0.180*** [0.0868,0.371] | |
| **Perceived burnout** | | | |
| No burnout | | | Ref. |
| Low burnout | | | 0.662* [0.474,0.923] |
| High burnout | | | 0.383*** [0.252,0.583] |
| *Control variables* | | | |
| **Job satisfaction prior to pandemic** | | | |
| Very dissatisfied | Ref. | Ref. | Ref. |
| Dissatisfied | 2.003 [0.738,5.433] | 1.987 [0.671,5.888] | 2.031 [0.684,6.030] |
| Satisfied | 14.92*** [5.641,39.45] | 16.32*** [5.672,46.97] | 18.21*** [6.310,52.54] |
| Very satisfied | 58.71*** [21.23,162.3] | 72.30*** [23.88,218.9] | 81.41*** [26.83,247.0] |
| **Perceived management appreciation** | | | |
| Not at all appreciative | Ref. | Ref. | Ref. |
| Somewhat appreciative | 2.884*** [1.823,4.564] | 2.931*** [1.801,4.771] | 2.313*** [1.409,3.798] |
| Appreciative | 4.990*** [2.957,8.421] | 4.818*** [2.750,8.441] | 3.956*** [2.235,7.003] |
| Very appreciative | 4.926*** [2.481,9.783] | 4.906*** [2.382,10.11] | 3.898*** [1.877,8.097] |
| **Fear of infection** | | | |
| Not fearful | Ref. | Ref. | Ref. |
| A little fearful | 0.665* [0.452,0.980] | 0.738 [0.490,1.112] | 0.670 [0.444,1.010] |
| Fearful | 0.598* [0.389,0.918] | 0.584* [0.370,0.922] | 0.567* [0.359,0.896] |
| Very fearful | 0.228*** [0.141,0.369] | 0.262*** [0.159,0.434] | 0.235*** [0.141,0.393] |
| **Management communication** | | | |
| Very poor | Ref. | Ref. | Ref. |
| Poor | 1.906* [1.162,3.126] | 2.241** [1.315,3.819] | 2.898*** [1.691,4.967] |
| Good | 2.182** [1.295,3.678] | 3.180*** [1.835,5.511] | 4.021*** [2.300,7.028] |
| Very good | 2.374* [1.149,4.908] | 3.617*** [1.694,7.726] | 4.292*** [1.992,9.244] |
| **Support from family** | | | |
| Not at all supportive | Ref. | Ref. | Ref. |
| A little supportive | 1.031 [0.549,1.936] | 0.766 [0.381,1.538] | 0.915 [0.459,1.825] |
| Supportive | 1.708 [0.929,3.142] | 1.218 [0.619,2.395] | 1.402 [0.717,2.740] |
| Very supportive | 1.961* [1.040,3.699] | 1.444 [0.717,2.907] | 1.613 [0.805,3.233] |
| **Gender** | | | |
| Male | Ref. | Ref. | Ref. |
| Female | 0.948 [0.724,1.243] | 0.893 [0.669,1.194] | 0.945 [0.703,1.269] |
| **Age** | | | |
| < 30 | Ref. | Ref. | Ref. |

*(Continued)*

**Table 3.** (Continued)

|  | Model I | Model II | Model III |
|---|---|---|---|
|  | IV: Perceived Preparedness | IV: Perceived Stress | IV: Perceived Burnout |
| 30–39 | 1.093 [0.782,1.528] | 1.086 [0.760,1.551] | 1.104 [0.771,1.581] |
| 40+ | 1.089 [0.714,1.661] | 1.194 [0.762,1.872] | 1.266 [0.803,1.998] |
| **Marital status** |  |  |  |
| Single | Ref. | Ref. | Ref. |
| Currently married | 0.812 [0.583,1.130] | 0.694* [0.488,0.987] | 0.637* [0.447,0.906] |
| Formerly married (divorced/widowed) | 0.743 [0.321,1.719] | 0.625 [0.260,1.504] | 0.504 [0.211,1.205] |
| **Position** |  |  |  |
| Doctor | Ref. | Ref. | Ref. |
| Nurse/related | 1.098 [0.770,1.566] | 1.164 [0.798,1.698] | 1.106 [0.755,1.620] |
| Other | 1.304 [0.846,2.010] | 1.235 [0.783,1.949] | 1.204 [0.759,1.908] |
| **Health facility type** |  |  |  |
| Teaching hospital | Ref. | Ref. | Ref. |
| Other government facility | 0.919 [0.657,1.284] | 0.919 [0.644,1.311] | 0.859 [0.601,1.228] |
| Private/Missions | 0.819 [0.518,1.295] | 0.898 [0.555,1.454] | 0.896 [0.552,1.456] |
| **country/phases** |  |  |  |
| Ghana Phase 1 | 0.743* [0.557,0.992] | 0.632** [0.466,0.859] | 0.620** [0.456,0.843] |
| Ghana Phase 2 | Ref. | Ref. | Ref. |
| Kenya | 0.801 [0.512,1.254] | 0.751 [0.465,1.215] | 0.804 [0.494,1.308] |
| Pseudo R-squared | 0.244 | 0.258 | 0.258 |
| Pseudo R-squared (excluding prior job satisfaction from the model) | 0.154 | 0.162 | 0.156 |
| Sample size | 1009 | 897 | 887 |

Exponentiated coefficients; 95% confidence intervals in brackets

* $p < 0.05$,

** $p < 0.01$,

*** $p < 0.001$.

## Discussion

In this multi-phased cross-sectional study of HCWs in Ghana and Kenya, we found that more than a third of providers in Ghana and Kenya were dissatisfied with their jobs during the pandemic. Reported levels of job dissatisfaction during the pandemic were greater than prior to the pandemic. But job dissatisfaction was higher in the early phase of the pandemic than in the later. There were, however, no significant differences between the providers in Kenya and Ghana around the same period in the later phase. Additionally, over two thirds of providers had low perceived preparedness, moderate to high stress, and low to high burnout. Consistent with our hypothesis, we found that higher perceived preparedness was associated with higher satisfaction, while high stress and burnout were associated with lower satisfaction. The effect of satisfaction mediated by stress and burnout was however not statistically significant. Other factors associated with current satisfaction were satisfaction prior to the pandemic, perceived appreciation and communication from management, and support of family. Fear of infection was associated with lower satisfaction.

To our knowledge, this is the first study to examine perceived preparedness and job satisfaction among HCWs in SSA during the COVID-19 pandemic. Our findings are, however, consistent with what is known about job satisfaction, broadly, and during the pandemic. Prior studies have found that job-related stress, perceived inadequate preparedness, and fear of

**Table 4. Multivariable model showing the association between perceived preparedness and current job satisfaction, controlling for perceived burnout, stress, and other socio-demographic factors.**

| | Model I | Model II | Model III |
|---|---|---|---|
| *Independent variable* | IV: Perceived Preparedness | IV: Perceived Preparedness & Stress | IV: Perceived Preparedness & Burnout |
| | APOR [95% CI] | APOR [95% CI] | APOR [95% CI] |
| **Perceived preparedness** | | | |
| Not at all prepared | Ref. | Ref. | Ref. |
| A little prepared | 1.679* [1.052,2.678] | 1.543 [0.918,2.594] | 1.531 [0.903,2.597] |
| Prepared | 2.831*** [1.657,4.838] | 2.776*** [1.533,5.029] | 2.638** [1.441,4.829] |
| **Perceived stress** | | | |
| Low stress | | Ref. | |
| Moderate stress | | 0.529*** [0.379,0.738] | |
| High stress | | 0.183*** [0.0882,0.379] | |
| **Perceived burnout** | | | |
| No burnout | | | Ref. |
| Low burnout | | | 0.712* [0.509,0.997] |
| High burnout | | | 0.424*** [0.278,0.648] |
| *Control variable* | | | |
| **Job satisfaction prior to pandemic** | | | |
| Very dissatisfied | Ref. | Ref. | Ref. |
| Dissatisfied | 2.003 [0.738,5.433] | 2.008 [0.667,6.049] | 2.022 [0.673,6.080] |
| Satisfied | 14.92*** [5.641,39.45] | 16.10*** [5.502,47.10] | 17.58*** [6.015,51.38] |
| Very satisfied | 58.71*** [21.23,162.3] | 65.95*** [21.41,203.1] | 74.11*** [24.08,228.0] |
| **Perceived management appreciation** | | | |
| Not at all appreciative | Ref. | Ref. | Ref. |
| Somewhat appreciative | 2.884*** [1.823,4.564] | 3.001*** [1.841,4.892] | 2.391*** [1.453,3.934] |
| Appreciative | 4.990*** [2.957,8.421] | 4.619*** [2.630,8.113] | 3.831*** [2.158,6.800] |
| Very appreciative | 4.926*** [2.481,9.783] | 4.438*** [2.141,9.203] | 3.634*** [1.739,7.591] |
| **Fear of infection** | | | |
| Not fearful | Ref. | Ref. | Ref. |
| A little fearful | 0.665* [0.452,0.980] | 0.792 [0.523,1.202] | 0.712 [0.469,1.079] |
| Fearful | 0.598* [0.389,0.918] | 0.645 [0.406,1.025] | 0.615* [0.388,0.977] |
| Very fearful | 0.228*** [0.141,0.369] | 0.297*** [0.177,0.498] | 0.256*** [0.152,0.432] |
| **Management communication** | | | |
| Very poor | Ref. | Ref. | Ref. |
| Poor | 1.906* [1.162,3.126] | 2.087** [1.219,3.573] | 2.694*** [1.563,4.644] |
| Good | 2.182** [1.295,3.678] | 2.533** [1.438,4.463] | 3.230*** [1.816,5.746] |
| Very good | 2.374* [1.149,4.908] | 2.552* [1.162,5.607] | 3.013** [1.360,6.675] |
| **Support from family** | | | |
| Not at all supportive | Ref. | Ref. | Ref. |
| A little supportive | 1.031 [0.549,1.936] | 0.758 [0.376,1.529] | 0.909 [0.454,1.820] |
| Supportive | 1.708 [0.929,3.142] | 1.168 [0.591,2.310] | 1.360 [0.692,2.672] |
| Very supportive | 1.961* [1.040,3.699] | 1.290 [0.637,2.612] | 1.472 [0.731,2.965] |
| **Gender** | | | |
| Male | Ref. | Ref. | Ref. |
| Female | 0.948 [0.724,1.243] | 0.914 [0.682,1.225] | 0.959 [0.713,1.291] |
| **Age** | | | |
| < 30 | Ref. | Ref. | Ref. |
| 30–39 | 1.093 [0.782,1.528] | 1.046 [0.730,1.500] | 1.062 [0.739,1.526] |

(*Continued*)

**Table 4.** (Continued)

|  | Model I | Model II | Model III |
|---|---|---|---|
| 40+ | 1.089 [0.714,1.661] | 1.081 [0.686,1.705] | 1.146 [0.722,1.818] |
| **Marital status** |  |  |  |
| Single | Ref. | Ref. | Ref. |
| Currently married | 0.812 [0.583,1.130] | 0.741 [0.518,1.058] | 0.676* [0.472,0.967] |
| Formerly married (divorced/widowed) | 0.743 [0.321,1.719] | 0.632 [0.259,1.538] | 0.509 [0.211,1.229] |
| **Position** |  |  |  |
| Doctor | Ref. | Ref. | Ref. |
| Nurse/related | 1.098 [0.770,1.566] | 1.183 [0.808,1.733] | 1.117 [0.760,1.641] |
| Other | 1.304 [0.846,2.010] | 1.290 [0.814,2.043] | 1.257 [0.790,1.999] |
| **Health facility type** |  |  |  |
| Teaching hospital | Ref. | Ref. | Ref. |
| Other government facility | 0.919 [0.657,1.284] | 0.992 [0.693,1.421] | 0.930 [0.648,1.335] |
| Private/Missions | 0.819 [0.518,1.295] | 0.949 [0.581,1.549] | 0.957 [0.584,1.568] |
| **country/phases** |  |  |  |
| Ghana Phase 1 | 0.743* [0.557,0.992] | 0.709* [0.517,0.972] | 0.693* [0.505,0.951] |
| Ghana Phase 2 | Ref. | Ref. | Ref. |
| Kenya | 0.801 [0.512,1.254] | 0.804 [0.494,1.310] | 0.867 [0.529,1.421] |
| Pseudo R-squared | 0.244 | 0.265 | 0.263 |
| Pseudo R-squared (excluded prior job satisfaction from the model) | 0.154 | 0.175 | 0.168 |
| Sample size | 1009 | 890 | 880 |

Exponentiated coefficients; 95% confidence intervals in brackets.

* $p < 0.05$,

** $p < 0.01$,

*** $p < 0.001$.

COVID-19 infection contribute to lower levels of job satisfaction [3, 53]. Increased risk of exposure to COVID-19 due to increasing cases has contributed to providers' fear of becoming infected and infecting their friends and family [53–56]. As a result, providers experience higher levels of psychological stress, burnout and anxiety, as well as decreased job satisfaction [34, 53, 56, 57].

A qualitative study conducted in Jordan found that shortages of PPEs and decreased HCW preparedness, negatively impacted psychological wellbeing and ultimately contributed to lower levels of job satisfaction [53]. HCWs in healthcare facilities with adequate provision of resources (PPEs and healthcare personnel) on the other hand, reported higher levels of job satisfaction [53]. Concurrent with our findings, bureaucracy, low salaries, and strained workplace relationships between management and staff have been reported in the literature as major factors contributing to job satisfaction in the COVID-19 context [3]. Previous studies on factors that influence job satisfaction have also shown that improved staff relations and organizational structure, adequate staffing of healthcare personnel, a safe working environment, and appreciation by management are effective strategies for improving job satisfaction [58–60].

There are, however, a few inconsistencies with the literature. For example, a study conducted in Spain found that nurses working during the COVID-19 pandemic had high levels of job satisfaction [61]. The researchers theorized that this could be due to nurses having a high level of resilience and a keen awareness of their role and importance in combatting the

pandemic [61]. This difference could also be attributed to the nurses working in high resource settings and, therefore, not experiencing the challenges of working in limited-resource settings, such as personnel shortages and less developed clinical infrastructure.

Our finding on the relationship between stress and burnout and poor job satisfactions is also expected. For example, a mixed-methods study conducted in Jordan found that burnout was a significant predictor of lower levels of job satisfaction [62]. However, we had anticipated that a significant proportion of the effect of perceived preparedness on job satisfaction would be mediated by stress and burnout, given that low perceived preparedness increases the risk of high stress and burnout, which could lead to low job satisfaction. A possible reason for the non-significant mediated effect is that poor preparedness has implications for job satisfaction even if it does not lead to high stress and burnout. Another possible reason is the bi-directional nature of stress and burnout and satisfaction, which cannot be assessed with cross-sectional data.

Another key finding is that providers who participated in the survey in the later phase of the pandemic (Ghana phase 2 and Kenya) had higher satisfaction than those who participated in earlier phase. One reason for this is that, in the initial period of the pandemic, less was known about the disease, there were fewer guidelines for management, many facilities felt unprepared to deal with it, and there was little promise of vaccines or effective treatments. There was thus more panic and desperation among providers, which would have influenced satisfaction levels. However, with time, guidelines became available, many providers received training, and facilities were able to put in place measures to increase preparedness such as have more PPES available, which could account for improvements in satisfaction levels. In addition, hopes for vaccines and treatment were much higher in the second phase, which likely improved providers' satisfaction with their jobs.

It must be noted however that, the different incentives (raffle in second but not first phase) could have introduced some bias, such as more motivated people participating in the first phase when there were no incentives and some less motivated people also participating in the second phase because of the incentive. This could potentially underestimate the difference in satisfaction between the two phases if such motivation is associated with satisfaction. But we are unable to confirm this. However, given the similar satisfaction rate for Ghana phase 2 and Kenya, where no incentive was offered, the incentive may not have influenced the results very much. In addition, since the completion of our survey, both Ghana and Kenya have experienced a second wave of infection due to decreasing adherence to preventive measures and low vaccination rates from poor vaccine supply and vaccine hesitancy. It is thus possible that this could again lower job satisfaction among healthcare workers.

Despite the differences in the HCW-patient ratio in Kenya and Ghana, we did not find significant differences in job satisfaction among the HCWs in the two countries during the same period in the pandemic. This is likely because the differences in HCW-patient ratios are accounted for by burnout, which was significantly associated with job satisfaction in both countries. The predictors of job satisfaction were also consistent, although the confidence intervals were much wider with the Kenya sample because of the smaller sample size. The significant associations with the various predictors such as perceived preparedness, appreciation from management, and support from their family suggest these factors have some independent effects on job satisfaction regardless of the HCW-patient ratio.

Although previous studies have reported gender differences in job satisfaction, with women more likely to be happy with their jobs [63, 64], we did not find significant differences in job satisfaction by gender. This is likely because in the context of the pandemic, the more proximal factors such as perceived preparedness, stress, burnout, appreciation, and support are more important than the demographic factors. These factors are also likely associated with work

orientation, which has been shown to explain away the effect of gender on job satisfaction [63]. A similar explanation will apply to the lack of significant difference by provider and facility type.

## Implications

Our findings contribute data on frontline workers and has implications for the pandemic response in Africa, particularly given the effects of job satisfaction on job performance, commitment, absenteeism, retention, and turnover rates. Importantly, HCW concerns about preparedness, support, and mental health must be addressed, given the evidence that these factors shape job satisfaction among HCWs. Interventions to address stress and burnout are particularly important and needed. Programs like workplace mindfulness training, stress management initiatives, and peer support have been found to mitigate the effects of stress and burnout [65]. However, organizational and health system changes are required for sustained change [65]. The governments of Ghana and Kenya, as well as hospital management in each respective country must therefore take meaningful steps to support HCWs nationally and locally. For example, adequate communication from the government and hospital management, trainings, increased and timely pay, incentives, and workforce expansion may increase the capacity, confidence, and morale of HCWs in responding to the pandemic and improve their job satisfaction. Expression of appreciation from leadership could also make a big difference. Appreciation may include word of affirmation or gratitude as well as tangible gifts from management to staff. Communication from management to staff must be clear, consistent, respectful, and empathetic. Additionally, to understand and address the extent of and underlying causes of low job satisfaction, further research, particularly qualitative research, is needed on the various affective, evaluative, and behavioral components of job satisfaction and factors shaping these in the COVID-19 pandemic context.

## Limitations and strengths

The primary limitation of the study is that it is a cross-sectional design. Thus, we are unable to make causal inference based on the temporal order of events. Theoretically, it makes sense to assume that low perceived preparedness for the pandemic will lead to poor job satisfaction. However, a bi-directional relationship between satisfaction and stress and burnout is plausible. Additionally, unlike preparedness, stress, and burnout, which were based on composite scores, we included only a single question on overall satisfaction to keep the survey manageable. Although this question captures people's emotional response about their job, it may not capture the full range of feelings and beliefs about their jobs, potentially underestimating the degree of job dissatisfaction. Also, as with all self-reported data, recall and social desirability bias are limitations, although not a major factor as many of the questions did not require long recall periods. Further, completing the survey anonymously and online mitigates social desirability bias. Another limitation relates to generalizability given that this was a convenient sample of HCWs. We attempted to improve this by using various recruitment strategies as described in the methods. This was, however, more successful in Ghana than in Kenya, which was included in only the second phase, and where we had less success in recruitment leading to a relatively smaller sample. Offering an incentive in phase two likely helped with recruitment in Ghana but could have contributed to selection bias. Further, because of the relatively small sample size for Kenya, and the lack of distinction made in the survey for HCWs in training, we did not conduct sub-group analyses for different categories of healthcare works, including physicians in training, who may be experiencing higher levels of distress. Other limitations are the pre-existing differences between Kenya and Ghana, and the differences in

sample size and incentives in different countries and time frames. But as discussed above the effects of these differences were likely minimal due to the consistency of the results in the stratified analysis. Further, this is among the few studies in Africa to examine issues of preparedness, stress, burnout, and job satisfaction during the COVID-19 pandemic. The findings will contribute to addressing the challenges in Africa's response to this and future pandemics.

## Conclusions

Many HCWs in Ghana and Kenya are dissatisfied with their jobs and this has increased with the COVID-19 pandemic. For HCWs, the year has been particularly hard given their role as frontline workers in the pandemic response while harboring fears for their own lives and that of their families. This has led to high stress and burnout among HCWs. For providers in low-resource settings, there is the added burden of low perceived preparedness. All these factors lead to poor job satisfaction, which has implications for performance, retention, and quality of care in the healthcare sector. Given the already precarious position of the healthcare workforce in Africa, it is important that efforts are put in place to improve job satisfaction among HCWs, a group whose labor and expertise are central to curbing the pandemic. This should include efforts to increase preparedness, including training, and making available PPEs, isolation wards, and clear guidelines for prevention and management of COVID-19. Beyond that, we need health systems strengthening activities that will decrease stress and burnout such as increasing the number and skill of HCWs, as well as making improvements to infrastructure, equipment, medicines, and supplies for healthcare provision. Our findings also suggest simpler approaches such as appreciation and effective communication from management, in addition to support from families could help improve job satisfaction. To enable Africa to contain the pandemic and prepare for future pandemics, HCWs must be motivated and supported to be invested in their jobs and improving job satisfaction through these efforts will be critical to ensuring this.

## Supporting information

**S1 Questionnaire. Study questionnaire.** Sections of study questionnaire relevant to this manuscript.
(PDF)

**S1 Dataset. Data underlying the results of the study.** Data file in STATA.
(DTA)

**S1 Table. All supplemental tables.**
(DOCX)

## Acknowledgments

We thank all healthcare providers who participated in the study and who helped in the survey dissemination.

## Author Contributions

**Conceptualization:** Patience A. Afulani, Jerry John Nutor, Raymond A. Aborigo, John Koku Awoonor-Williams.

**Data curation:** Patience A. Afulani, Jerry John Nutor, Akua O. Gyamerah, Joseph Musana, Raymond A. Aborigo, Osamuedeme Odiase, Monica Getahun, Linnet Ongeri, Hawa Malechi, Moses Obimbo Madadi, Benedicta Arhinful, John Koku Awoonor-Williams.

**Formal analysis:** Pascal Agbadi.

**Funding acquisition:** Patience A. Afulani, Jerry John Nutor.

**Investigation:** Patience A. Afulani, Jerry John Nutor, Joseph Musana, Raymond A. Aborigo, Linnet Ongeri, Hawa Malechi, Moses Obimbo Madadi, Benedicta Arhinful, John Koku Awoonor-Williams.

**Methodology:** Patience A. Afulani, Jerry John Nutor, Pascal Agbadi, Akua O. Gyamerah, Raymond A. Aborigo, John Koku Awoonor-Williams.

**Project administration:** Patience A. Afulani, Akua O. Gyamerah, Joseph Musana, Raymond A. Aborigo, Monica Getahun.

**Resources:** Patience A. Afulani, Jerry John Nutor.

**Supervision:** Patience A. Afulani, Raymond A. Aborigo.

**Validation:** Patience A. Afulani, Jerry John Nutor, Pascal Agbadi, Akua O. Gyamerah, Joseph Musana, Raymond A. Aborigo, Osamuedeme Odiase, Monica Getahun, Linnet Ongeri, Hawa Malechi, Moses Obimbo Madadi, Benedicta Arhinful, John Koku Awoonor-Williams.

**Writing – original draft:** Patience A. Afulani, Jerry John Nutor, Pascal Agbadi, Akua O. Gyamerah, Raymond A. Aborigo, Osamuedeme Odiase, Ann Marie Kelly, John Koku Awoonor-Williams.

**Writing – review & editing:** Patience A. Afulani, Jerry John Nutor, Pascal Agbadi, Akua O. Gyamerah, Joseph Musana, Raymond A. Aborigo, Osamuedeme Odiase, Monica Getahun, Linnet Ongeri, Hawa Malechi, Moses Obimbo Madadi, Benedicta Arhinful, John Koku Awoonor-Williams.

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
