## [Decision Letter · Decision Letter 0]

9 Aug 2021

 PGPH-D-21-00322 Job satisfaction among healthcare workers in Ghana and Kenya during the COVID-19 pandemic: role of perceived preparedness, stress, and burnout PLOS Global Public Health

Dear Dr. Afulani,

Thank you for submitting your manuscript to PLOS Global Public Health. After careful consideration, we feel that it has merit but does not fully meet PLOS Global Public Health’s publication criteria as it currently stands. Therefore, we invite you to submit a revised version of the manuscript that addresses the points raised during the review process.

We look forward to receiving your revised manuscript.

Kind regards,

Kathleen Bachynski, PhD, MPH

Academic Editor

Journal Requirements:

Additional Editor Comments (if provided):

Thank you very much for submitting your manuscript to PLOS Global Health. The manuscript addresses an important area of research in global health and, as reviewer 1 highlighted, makes a novel contribution in this area of investigation. Reviewer 2 recommends some revisions, particularly incorporating some additional discussion of some of the findings and limitations of the study. Therefore, I invite you to respond to the reviewer’s comments and revise the manuscript.

In discussing the settings of Ghana and Kenya (in the methods section), you might also briefly note if there are any key differences worth highlighting in addition to the similarities you discuss regarding Covid-19 trends and constrained health systems in the two countries. For instance, you indicate that there are nearly four times as many nurses/midwives per 10,000 population in Ghana as compared to Kenya (42 per 10,000 compared to 8.3 per 10,000); does this difference have any potential relevance for job satisfaction among healthcare workers in the two countries? Generally, as Reviewer 2 suggests, some additional explicit discussion of the rationale for comparing healthcare workers in Ghana and Kenya, including differences in methodology between the countries (differences in time frame, sample size, use of a raffle incentive in Ghana but not in Kenya, etc.), and some additional discussion of the potential limitations associated with these differences that might affect how the findings are interpreted, would further strengthen the manuscript.

Reviewers' comments:

Reviewer's Responses to Questions

**Comments to the Author**

1. Does this manuscript meet PLOS Global Public Health’s publication criteria? Is the manuscript technically sound, and do the data support the conclusions? The manuscript must describe methodologically and ethically rigorous research with conclusions that are appropriately drawn based on the data presented.

Reviewer #1: Yes

Reviewer #2: Partly

2. Has the statistical analysis been performed appropriately and rigorously?

Reviewer #1: Yes

Reviewer #2: I don't know

3. Have the authors made all data underlying the findings in their manuscript fully available (please refer to the Data Availability Statement at the start of the manuscript PDF file)?

Reviewer #1: Yes

Reviewer #2: Yes

4. Is the manuscript presented in an intelligible fashion and written in standard English?

Reviewer #1: Yes

Reviewer #2: Yes

5. Review Comments to the Author

Reviewer #1: Dear Author (s)

I read the manuscript entitled “Job satisfaction among healthcare workers in Ghana and Kenya during the COVID-19 pandemic: role of perceived preparedness, stress, and burnout”,

Overall, this is a clear, detailed, and well-written manuscript. The introduction is relevant, and theory based. Sufficient information about the previous study findings is presented for readers to follow the present study rationale and procedures. The methods are generally appropriate. Overall, the results are clear. The authors make a systematic contribution to the research literature in this area of investigation. Overall, this is a high-quality manuscript that has implications for the theoretical basis.

Regards

Reviewer #2: Thank you for allowing me to review this interesting manuscript. Here are a few of my thoughts and suggestions:

a. I am unable to tell from the manuscript if there was difference between the different type of healthcare workers survey in both Ghana and Kenya. Were nurse more affected than physician or vice versa? Who makes up the allied healthcare workers? How physicians were in training as they tend to suffer more from psychological differences.

b. It is also unclear from the survey that all healthcare workers were involved with patient who had COVID-19. This would be useful to fully determine a solid relationship.

c. The study was conducted by mainly OBGYN department, who I am not sure are fully involved in COVID care, certainly at hospital in Kenya.

d. The offering of a raffle to the participants in Ghana would, I think, cause a bias in the responses. This should be a limitation. Alternatively, this population should be removed.

e. One of the questions that always comes up time and time again is the validation of the survey tools. Are the surveys used in this study validated in West and East Africa? Can the authors provide justification to the validations process or quote other studies that have used these tools?

f. The authors did not really provide a justification or explanation about why symptoms were worse in the first wave rather than the second phase of the study. This would be key to help explain their findings. I would assume that as number of waves increase, the symptoms would worsen.

g. The number of participants in Kenya was only 128. This is a huge limitation and such low numbers cannot be generalized to the Kenya healthcare worker force. I wonder if removing the Kenya portion of the study would make this study stronger and more applicable.

h. The authors also need to explain why they did not find a difference between job satisfaction, gender and type of healthcare facility. How many facilities in Kenya were recruited and are these private, governmental or other hospitals? Resources at different facilities differ and should be a limitation of the study as well.

i. I think the introductions could be improved better with additional information on what other studies have shown and the lack of data in this area. Also the discussion needs more work and I would request that authors to highlight clearly all their impactful findings and explain them with a justification.

j. The authors makes suggestion of how to mitigate their findings in the last sections of the manuscript. This needs clarity and in depth explanation of what they are exactly suggesting to be impactful to the readers especially in low resources areas. What are some of the suggested strategies for the administration to better appreciate staff and improve communication? This blanket statement might be more meaningful if we can adopt ideas implement elsewhere.

6. PLOS authors have the option to publish the peer review history of their article (what does this mean?). If published, this will include your full peer review and any attached files.

**Do you want your identity to be public for this peer review?** For information about this choice, including consent withdrawal, please see our Privacy Policy.

Reviewer #1: **Yes: **Saad Ahmed Ali jadoo

Reviewer #2: No

---

## [Editor Report · Decision Letter 1]

15 Sep 2021

Job satisfaction among healthcare workers in Ghana and Kenya during the COVID-19 pandemic: role of perceived preparedness, stress, and burnout

PGPH-D-21-00322R1

Dear Dr. Afulani,

We're pleased to inform you that your manuscript has been judged scientifically suitable for publication and will be formally accepted for publication once it meets all outstanding technical requirements.

Within one week, you'll receive an e-mail detailing the required amendments. When these have been addressed, you'll receive a formal acceptance letter and your manuscript will be scheduled for publication.

An invoice for payment will follow shortly after the formal acceptance. To ensure an efficient process, please log into Editorial Manager at https://www.editorialmanager.com/pgph/ click the 'Update My Information' link at the top of the page, and double check that your user information is up-to-date. If you have any billing related questions, please contact our Author Billing department directly at authorbilling@plos.org.

Kind regards,

Kathleen Bachynski, PhD, MPH

Academic Editor

Additional Editor Comments (optional):

My thanks to the authors for their detailed, thoughtful responses to the reviewer and editorial comments. The revisions provide valuable additional context and further strengthened the manuscript

A very minor note for page 30 of the revision is that there appears to be a missing word in this sentence: "These factors are also likely associated work orientation..." This can be corrected in copy editing.

I'm pleased to accept this manuscript for publication in PLOS Global Public Health.